# Computational analysis of the deleterious non-synonymous single nucleotide polymorphisms (nsSNPs) in TYR gene impacting human tyrosinase protein and the protein stability

Wei Fan[1‡], Heng Li Ji[2‡], Mohibullah Kakar[3], Shabbir Ahmed[4]*, Hussah M. Alobaid[5], Yasmeen Shakir[6]*

1 Department of Laboratory Medicine, Huaian Cancer Hospital, Huaian, Jiangsu, China, 2 Nephrology Department, Huaian Cancer Hospital, Huaian, Jiangsu, China, 3 Faculty of Marine Sciences Lasbela University of Agriculture Water and Marine sciences (LUAWMS), Uthal, Balochistan, Pakistan, 4 Faculty of Animal Husbandry and Veterinary Science, Sindh Agriculture University, Tandojam, Pakistan, 5 Zoology Department, College of Science, King Saud University, Riyadh, Saudi Arabia, 6 Department of Biochemistry, Hazara University, Mansehra, Khyber Pakhtunkhwa, Pakistan

‡ WF and HLJ contributed equally to this work and share the first authorship.
* shabbirch983@gmail.com (SA); yasmeenshakir@hotmail.com (YS)

**Data Availability Statement:** All relevant data are within the paper and its Supporting information files.

## Abstract

Tyrosinase, a copper-containing oxidase, plays a vital role in the melanin biosynthesis pathway. Mutations in the tyrosinase gene can disrupt the hydroxylation of tyrosine, leading to decreased production of 3,4-dihydroxyphenylalanine (DOPA). Consequently, this impairs the subsequent formation of dopaquinone, a key precursor in melanin pigment synthesis. This study aimed to identify the deleterious non-synonymous single nucleotide polymorphisms (nsSNPs) within the TYR gene that exert an influence on the human TYR protein. Additionally, we evaluated the impact of 10 FDA-approved drugs on the protein stability of mutated structures, exploring the potential for inhibitory pharmaceutical interventions. Through various bioinformatics tools, we detected 47900 nsSNPs, particularly K142M, I151N, M179R, S184L, L189P, and C321R, which were found to be the most deleterious variants, decreasing the protein stability. These drugs (Sapropterin, Azelaic Acid, Menobenzone, Levodopda, Mequinol, Arbutin, Hexylresorcinol, Artenimol, Alloin and Curcumin) interacted with the binding sites in four mutant models K142M, I151N, M179R, and S184L proving that these ligands directly bind with the active site of mutant tyrosinase protein to inhibit it's working. On the other hand, two mutant models L189P and C321R did not show any binding site residue interaction with any ligands. In conclusion, this in-silico analysis of deleterious nsSNPs in the TYR gene, coupled with the evaluation of ligands/drugs on mutated tyrosinase structures not only advances our understanding of molecular variations but also highlights promising pathways for targeted inhibitory interventions in the intricate network of melanin biosynthesis.

**Funding:** This work was funded by the Researchers Supporting Project number (RSP2023R97), King Saud University, Riyadh, Saudi Arabia. The funders had no role in study design, data collection and analysis, decision to publish, or preparation of the manuscript.

## Introduction

Pigmentation is a complex biological process that imparts color to an organism's skin, hair, feathers, and various structures through the presence of pigments [1]. These pigments, which possess the ability to absorb and reflect specific light wavelengths, contribute diverse types to the surrounding tissues. Among these pigments, melanins stand out as the most prevalent in animals and are produced by specialized cells known as melanocytes [2, 3]. Melanin exhibits a spectrum of colors, ranging from black to brown to yellow, playing a crucial role in determining the coloration of an animal's skin, hair, and eyes [4]. The orchestration of pigmentation involves a network of genes, such as TYR, MC1R, OCA2, SLC45A2, ASIP, and KIT. The KIT gene is particularly involved in the development of melanocytes, the cells responsible for producing pigments [5, 6]. On the other hand, MC1R and SLC45A2 play pivotal roles in the production of melanin pigment. ASIP and OCA2 contribute to the regulation and transportation of melanin, respectively, adding layers of complexity to the fine-tuned process of pigmentation.

Tyrosinase, a crucial copper-containing oxidase, plays a vital role in the melanin pigment production, dictating the vibrant colors of skin, hair, and eyes in both human and animals [1, 4, 7]. Serving as a linchpin in various biosynthetic pathways, this multifaceted enzyme catalyzes the hydroxylation of tyrosine, initiating the synthesis of 3,4-dihydroxyphenylalanine (DOPA) [8, 9]. This considered as a rate-determining step, propels the subsequent conversion of DOPA into dopaquinone, which further transforms into eumelanin and pheomelanin—distinct forms of melanin responsible for diverse pigmentation. Beyond its primary role in melanogenesis, tyrosinase showcases versatility by contributing to neurotransmitter synthesis, insect cuticle development, and the production of catecholamines and thyroid hormones [10, 11]. Moreover, the variations in TYR gene are associated with various genetic disorders affecting pigmentation, such as oculocutaneous albinism type 1 (OCA1) and melanoma [12]. OCA1 is a rare autosomal recessive disorder characterized by the absence or significant reduction of melanin, leading to vision problems, photophobia, and an increased susceptibility to skin cancer due to reduced protection against UV radiation [13]. In addition, the variations in this gene is implicated in the development of certain types of melanoma, a form of skin cancer arising from melanocytes [13]. Therefore, this gene has key role in pigmentation.

The tyrosinase gene spans a total length of 117,885 base pairs and is situated at the 11q14.3 locus, having six exons. The protein sequence encoded by tyrosinase consists of 529 amino acids, with a molecular mass of 60,393 Da [14]. Within the genetic variations, SNPs play a significant role, occurring at single positions in DNA sequences in both coding and noncoding regions of proteins [15]. Notably, SNPs in the coding region can induce alterations in the amino acid sequence, thereby introducing structural and functional modifications to the normal protein [16]. There are three primary types of SNPs in the coding region: synonymous SNPs (sSNPs), which maintain the amino acid sequence; nonsynonymous SNPs (nsSNPs), causing changes in the protein sequence and subsequent structural modifications; and nonsense SNPs, which generate premature stop codons, resulting in the production of non-functional truncated proteins [17]. A study indicates that approximately 50% of disease-causing variants in TYR gene are nsSNPs [18]. The impact of an SNP on the structure and function of a protein is contingent on the specific amino acid affected and its location within the protein. Currently, more than 600 million SNPs have been identified in the human genome, with a substantial proportion residing in the coding regions of proteins [19].

Various synthetic and natural compounds may use to treat melanoma and skin diseases via target the TYR gene [20–22]. Sapropterin, a synthetic drug, treats phenylketonuria by influencing the metabolism of phenylalanine [23]. Azelaic Acid, used topically, reduces inflammation

and fights bacteria in skin conditions like acne [24]. Menobenzone is a chemical in some sunscreens, absorbing UV radiation [25]. Levodopa, used for Parkinson's disease, is a precursor to dopamine [26]. Mequinol and arbutin are depigmenting agents used in skincare, while hexyl-resorcinol has antiseptic properties [27]. Artenimol treats malaria, alloin in aloe vera has laxative properties, and curcumin in turmeric is studied for its anti-inflammatory and antioxidant benefits [28]. In addition, the imperative to expedite the analysis of the huge number of variations in diverse genomes is being studied using different bioinformatics algorithms and tools [29]. These user-friendly resources play a crucial role in efficiently screening and functionally assessing critical nsSNPs, which exert influence over structural conformations. Therefore, this study aimed to identify deleterious nsSNPs within the TYR gene and understand their impact on the human TYR protein. Additionally, the study meticulously explores the effects of ligands on the stability of mutated protein structures, with a focus on therapeutic interventions possessing inhibitory potential.

## Methodology

### Prediction of nsSNPs in the TYR gene and protein

In the exploration of the TYR gene, 47,900 Single Nucleotide Polymorphisms (SNPs) have been meticulously investigated, revealing 651 non-synonymous variations. This comprehensive dataset, encompassing RS_ID, chromosome number, position, and variants, was meticulously sourced from the National Center for Biotechnology Information (NCBI) database. Concurrently, the protein sequence was retrieved from UniProt KB (ID: P14679). To discern the noteworthy impact of non-synonymous SNPs (nsSNPs) within the human TYR gene, an array of nine in-silico tools was judiciously employed. These tools include SIFT (Sorting Intolerant from Tolerant), Polyphen2 (Polymorphism Phenotyping v2), CADD (Combined Annotation-Dependent Depletion), Condel (CONsensus DELeteriousness score), SNAP2, PANTHER (Protein Analysis Through Evolutionary Relationships), SNP&Go (Single Nucleotide Polymorphism database & Gene ontology), PhD-SNP (Predictor of human Deleterious Single Nucleotide Polymorphisms), and P Mut predictor (Mutation's Pathology Predictor). Each algorithm was executed using default settings, with specific cutoff values applied. Notably, the CADD tool identified variants with a Phred score exceeding 20, representing the top 1% from Genome-Wide Associated Studied Data. Other tools were employed with the following criteria: SIFT p-value < 0.05, PolyPhen p-value > 0.95, Condel score > 0.522, SNAP2 score > 20, PANTHER p-value > 0.5, SNP&GO p-value > 0.5, PhD-SNP p-value > 0.5, and P Mut p-value > 0.5 [30]. The selection criterion for further analysis involved filtering the most common deleterious nsSNPs verified by at least 8 out of the 9 tools. This approach ensures a robust and reliable identification of potentially impactful variations within the TYR gene, laying the foundation for further analysis.

### Impact of selected deleterious nsSNPs on the protein stability

Mu Pro (http://mupro.proteomics.ics.uci.edu/) and I-Mutant (http://folding.biofold.org/i-mutant/i-mutant2.0.html) tools were used to investigate the impact of selected deleterious nsSNPs on the protein stability. A decrease in the protein stability means that the mutation causes conformational alterations that may inhibit its normal functioning in biological pathways. The accuracy of the I-Mutant tool is 77% where the amino acid sequence and the mutations of residues and their respective positions are taken as input. Mu Pro, a machine learning tool, on the other hand, shows the differences in the protein state and strengths caused by amino acid variants. ΔΔG<0 is taken as the cutoff value for both tools, describing the mutation as damaging.

Consurf (http://ConSurf.tau.ac.il/), a web-based server, gives information on structurally buried and functionally exposed amino acids in a protein structure. It also helps in determining whether the mutation resides in the conserved region or not. The conservation scale ranges from 1 to 9, where a 1–3 score refers to the variable, a 4–6 score is associated with the average region, and a score of 7–9 on the scale shows a highly conserved region [31].

## Structural analysis of deleterious nsSNPs

Hope is a web-based server (https://www3.cmbi.umcn.nl/hope/) used to analyze the structural effects of amino acid mutations in protein sequences. It gives information about the hydrophobicity, size, and charge of wild-type and mutant structures. The most deleterious nsSNPs which decreased the protein stability, caused more hydrophobicity, changed the overall charge of the protein, and are present in the most conserved region of a sequence selected for further study [32]. Inter Pro, an online server (https://www.ebi.ac.uk/interpro/) was used to study the functional domains of the tyrosinase protein. Interpro classifies proteins into families and provides functional analysis by predicting the domains and biological sites [33].

## Molecular docking analysis

AF-P14679-F1-model_v4 model from the AlphaFold database was downloaded and mutant models were built through the FoldX suite. Previously, only one PDB complex (7RK7) of tyrosinase bound with TIL 1383i-TCR and human_class I MHC HLA-A2 has been reported, which has a short tyrosinase nonameric peptide (369–377 AA). Therefore, we searched for the complete structure of tyrosinase in the AlphaFold structure prediction server, an AI-based computational algorithm to predict possible 3D structures of unknown proteins. FoldX Suite version 5.0 (http://foldxsuite.crg.eu/) was used to build mutagen models and their energy minimization. PyMol was used for the visualization of native and mutant models [34]. Molecular docking studies were conducted to assess structural changes induced by mutations in tyrosinase. The STRING database (https://string-db.org/) identified proteins in melanin-production pathways associated with tyrosinase. FDA-approved drugs targeting tyrosinase for controlled melanin production and depigmentation, sourced from DrugBank (https://go.drugbank.com/drugs/DB16626) and HMDB (https://hmdb.ca), were selected as ligands. Using AutoDock Vina in PyRx-0.8, wild type and mutant protein models were docked with 20 chosen drugs within a search space of X:168.6175Å, Y:99.9621Å, and Z:104.3704Å. Discovery Studio Visual 2021 facilitated the visualization of docked complexes. Employing a semi-flexible docking protocol with specified torsional degrees of freedom, binding residues of the target protein were kept flexible while others remained rigid [30]. The top 5 ligands, based on minimum binding energies in the docked complexes, were selected for detailed analysis, showcasing their potential impact on tyrosinase function.

## Results

### Prediction of the nsSNPs in TYR gene

The human TYR gene contains a total of 47900 single nucleotide polymorphisms (SNPs). 45531 (95.054%) SNPs lay in the intronic region and 1359 (2.837%) SNPs lay in the UTR regions. 651(1.359%) SNPs were non-synonymous whereas, 259 (0.545) SNPs were synonymous. Other types of SNPs are Stop Gained 37 (0.077%), Splice Site 33 (0.068%), and Frameshift 30 (0.062%) as depicted in Fig 1.

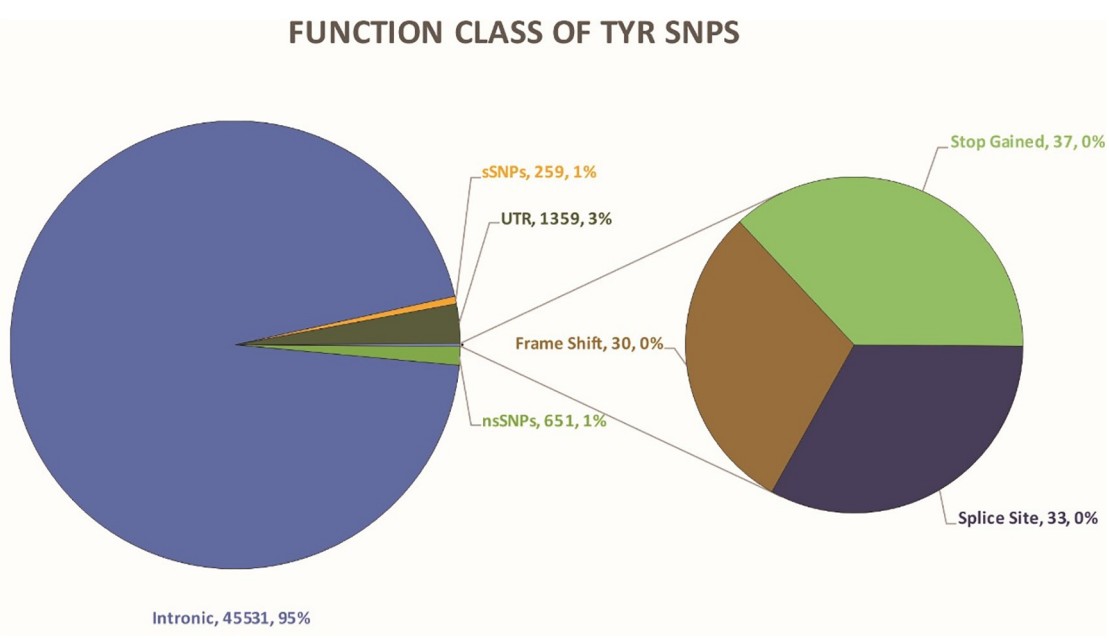

**Fig 1. Distribution of SNPs.**

## Prediction of the deleterious nsSNPs in the TYR protein and its stability

In the quest to identify deleterious and disease-causing non-synonymous single nucleotide polymorphisms (nsSNPs), 651 variants were analyzed. CADD identified 264 nsSNPs with a Phred score >20, SIFT revealed 234 variants with a P value <0.01, and Polyphen2 predicted 158 nsSNPs with a P value >0.995. Condel predicted 267 nsSNPs with a default score >5.22. SNAP2 and PROVEAN predicted a comprehensive list of 10,576 and 10,223 variants at the amino acid level, from which 651 variants of interest were filtered. SNP&GO predicted 524 nsSNPs, while the PhD SNP tool predicted 288 nsSNPs with a disease-causing effect and P value >0.5. P Mut tool identified 139 variants as disease-causing with P >0.5. Through rigorous filtering, 25 nsSNPs emerged as the most deleterious variants, unanimously identified by at least 8 out of 9 tools. Moreover, Mu Pro and I-Mutant revealed that 22 out of the 25 variants were predicted to decrease protein stability, with 3 variants supported by only one tool. The Consurf tool highlighted two structurally buried and four functionally exposed amino acids within the highly conserved region of the protein (Fig 2). Based on a detailed analysis (Table 1) pinpointed K142M, I151N, M179R, S184L, L189P, and C321R as the most deleterious variants, diminishing protein stability while residing in the highly conserved region of the tyrosinase protein is depicted in the S1 Table. All these five variants were supported by 11 out of 12 tools.

## Structural analysis of the deleterious nsSNPs in TYR protein

InterPro Scan revealed that the selected mutations are localized within the central tyrosinase copper-binding domain of the tyrosinase protein. Further analysis using the HOPE server provided the structural impact of mutations in the tyrosinase protein (Table 2). Notably, all mutants were found to impact the protein structure, with the exception of L189P. Additionally, GORIV analysis was employed to assess differences in amino acids within alpha helices, beta strands, and coils. Fig 3 visually represents the comparison of wild type and two mutant protein models (M179R and S184L), revealing their identical secondary structures marked with a

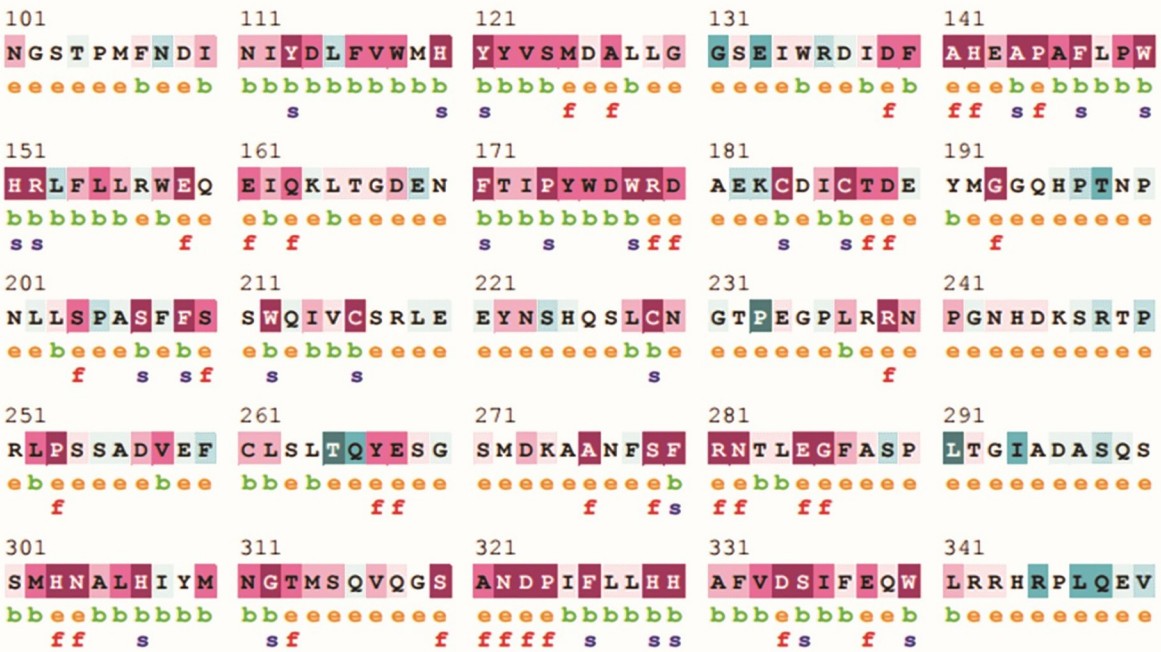

**Fig 2. ConSurf results for residues conservation.** Colors of ConSurf results showing the level of confidence for the sequence conservation. Light-blue color indicates the variables conservation at residue level, whereas dark purple color indicates highly conserved residues in Tyrosinase protein.

star. In contrast, the remaining four mutant models exhibited distinct secondary structures, as illustrated in the Fig 3.

## Molecular docking analysis

The predictive analysis of the tyrosinase protein's active pocket, including 28 pivotal amino acids (117Leu,118Leu, 119Val, 121Arg, 124Phe, 125Asp, 126Leu, 127Ser, 129Pro, 130Glu, 133Lys, 134Phe, 230Asn, 231Phe, 232Thr, 234Pro, 450Ser, 453Gln, 454Asp, 456Asp, 458Asp, 459Ser, 462Asp, 463Tyr, 465Lys, 466Ser, 467Tyr, 469Glu), were predicted. Fig 4A depicted the 3D structure of the native tyrosinase protein, while Fig 4B highlights the binding pocket in blue. Furthermore, Fig 5 predicted a visual representation of all mutations, with normal and mutant positions distinguished in blue and red, respectively. To probe potential therapeutic ways, 10 FDA-approved drugs—Sapropterin, Azelaic Acid, Menobenzone, Levodopa, Mequinol, Arbutin, Hexylresorcinol, Artenimol, Alloin, Curcumin—were subjected to docking studies with both native and six mutant protein models (Table 3). Remarkably, the Mutant model K142M demonstrated active site interactions with four ligands (Sapropterin, Artenimol, Alloin, and Curcumin), while the docked complexes of the I151N mutant model exhibited active site binding with five ligands (Menobenzone, Mequinol, Arbutin, Hexylresorcinol, and Artenimol). Similarly, the M179R mutant model displayed active site docking with five ligands (Sapropterin, Azelaic Acid, Menobenzone, Levodopa, and Hexylresorcinol). Conversely, S184L bound with Azelaic Acid, Levodopa, and Alloin through its active site. The native model displayed a more easily binding of the active site, interacting with select ligands such as Sapropterin, Mequinol, and Alloin. Among the total of 28 residues within the active site of the tyrosinase protein, only 15 residues actively participated in the binding process across various

**Table 1.** The functional consequences of nsSNPs and prediction of disease-associated nsSNPs in human TYR gene.

| Mutations | | | Deleterious Nature of nsSNPs | | | | | | | | | Effect on the Protein Stability | | Conserve Analysis |
|---|---|---|---|---|---|---|---|---|---|---|---|---|---|---|
| dbSNP rsID | Variant | AA Variant | SIFT | Polyphen | CADD | Condel | SNAP2 | PANTHER | SNP&GO | PhD SNP | P Mut | Mu Pro | I Mutant | Consurf |
| | | | Score <0.05 | Score >0.95 | Phred Score >20 | Score >0.522 | Score >20 | P Score >0.5 | P Score >0.5 | P Score >0.5 | P>0.5 | Effect | Effect | Score>7 |
| rs756049855 | A/T | D199V | 0 | 1 | 26.9 | 0.66 | 32 | 0.74 | 10 | 8 | 0.91 (93%) Disease | Decrease | Increase | 5,e |
| rs1013801316 | G/A | G346R | 0.01 | 0.994 | 35 | 0.7 | 29 | 0.85 | 10 | 9 | 0.69 (86%) Disease | Decrease | Decrease | 5,e |
| rs1395444792 | T/C | F392S | 0 | 0.996 | 31 | 0.71 | 27 | 0.74 | 10 | 7 | 0.78 (88%) Disease | Decrease | Decrease | 3,e |
| rs1287652457 | G/T | G436C | 0.02 | 0.974 | 24 | 0.58 | 28 | 0.57 | 10 | 8 | 0.70 (86%) Disease | Decrease | Decrease | 2,b |
| rs750027827 | G/T | C276F | 0 | 0.999 | 29 | 0.67 | 38 | 0.85 | 10 | 2 | 0.86 (91%) Disease | Decrease | Decrease | 5,e |
| rs200960909 | G/C | D42H | 0.01 | 0.999 | 23.2 | 0.64 | 23 | 0.74 | 10 | 5 | 0.58 (82%) Disease | Decrease | Decrease | 3,e |
| rs538081629 | G/A | A241T | 0.01 | 1 | 26.4 | 0.69 | -68 | 0.74 | 10 | 6 | 0.59 (82%) Disease | Decrease | Decrease | 7,e |
| rs759359525 | G/C | C244S | 0 | 0.999 | 26.6 | 0.63 | 29 | 0.85 | 10 | 3 | 0.78 (88%) Disease | Decrease | Decrease | 6,e |
| rs996525532 | T/C | C24R | 0 | 0.998 | 24.9 | 0.61 | 45 | 0.45 | 10 | 8 | 0.85 (91%) Disease | Decrease | Decrease | 7,b |
| rs373333305 | G/A | C24Y | 0 | 0.998 | 24.7 | 0.59 | 51 | 0.41 | 10 | 6 | 0.81 (89%) Disease | Decrease | Decrease | 7,b |
| rs938515275 | T/C | C321R | 0 | 0.999 | 26.1 | 0.63 | 39 | 0.57 | 10 | 1 | 0.60 (83%) Disease | Decrease | Decrease | 9,e,f |
| rs777884034 | G/T | D148Y | 0 | 0.841 | 25.4 | 0.64 | 64 | 0.85 | 10 | 8 | 0.89 (92%) Disease | Decrease | Decrease | 6,b |
| rs1204511442 | G/T | G101V | 0 | 0.994 | 26 | 0.66 | 49 | 0.46 | 10 | 6 | 0.87 (91%) Disease | Decrease | Decrease | 7,e |
| rs764905692 | G/C | G106A | 0.01 | 0.885 | 25.4 | 0.64 | 33 | 0.85 | 10 | 8 | 0.85 (91%) Disease | Decrease | Decrease | 5,e |
| rs1463152051 | G/T | G154W | 0.01 | 0.886 | 25.6 | 0.65 | 89 | 0.74 | 10 | 9 | 0.89 (92%) Disease | Decrease | Decrease | 6,b |
| rs747095957 | T/A | I151N | 0 | 0.998 | 26.3 | 0.68 | 21 | 0.48 | 10 | 9 | 0.86 (91%) Disease | Decrease | Decrease | 9,b,s |

(*Continued*)

**Table 1.** (Continued)

| Mutations | | | Deleterious Nature of nsSNPs | | | | | | | | | Effect on the Protein Stability | | Conserve Analysis |
|---|---|---|---|---|---|---|---|---|---|---|---|---|---|---|
| dbSNP rsID | Variant | AA Variant | SIFT | Polyphen | CADD | Condel | SNAP2 | PANTHER | SNP&GO | PhD SNP | P Mut | Mu Pro | I Mutant | Consurf |
| | | | Score <0.05 | Score >0.95 | Phred Score >20 | Score >0.522 | Score >20 | P Score >0.5 | P Score >0.5 | P Score >0.5 | P>0.5 | Effect | Effect | Score>7 |
| rs754250982 | A/T | K142M | 0 | 0.998 | 26.3 | 0.68 | 30 | 0.54 | 10 | 6 | 0.87 (91%) Disease | Decrease | Increase | 9,e,f |
| rs367963483 | T/G | L138R | 0 | 0.993 | 27.2 | 0.67 | 40 | 0.47 | 10 | 6 | 0.91 (93%) Disease | Decrease | Decrease | 6,b |
| rs1212067038 | T/C | L189P | 0 | 1 | 27.3 | 0.67 | 47 | 0.46 | 10 | 8 | 0.79 (89%) Disease | Decrease | Decrease | 8,e,f |
| rs769645029 | T/G | M179R | 0.02 | 0.953 | 24.9 | 0.66 | 38 | 0.74 | 10 | 2 | 0.77 (88%) Disease | Decrease | Decrease | 9,e,f |
| rs367543066 | C/T | S184L | 0.32 | 0.987 | 23.8 | 0.61 | 33 | 0.74 | 10 | 7 | 0.82 (90%) Disease | Decrease | Decrease | 9,b,s |
| rs1287735969 | G/T | W238L | 0.01 | 0.999 | 28.5 | 0.72 | 31 | 0.44 | 10 | 9 | 0.88 (92%) Disease | Decrease | Decrease | 5,e |
| rs748052034 | G/C | W475C | 0.01 | 0.996 | 29.2 | 0.76 | 56 | 0.46 | 10 | 6 | 0.84 (90%) Disease | Decrease | Decrease | 1,b |
| rs547058090 | A/G | Y173C | 0 | 1 | 26.8 | 0.74 | 36 | 0.41 | 10 | 7 | 0.91 (93%) Disease | Decrease | Decrease | 7,b |
| rs768206729 | G/C | G485R | 0.01 | 0.999 | 24.8 | 0.63 | 25 | 0.74 | 10 | 6 | 0.42 (85%) Neutral | Increase | Decrease | 1,b |

**Table 2. The structural consequences of deleterious nsSNPs in human TYR gene and domain analysis.**

| Mutation | HOPE server | | | | InterProScan |
|---|---|---|---|---|---|
| | More Hydrophobic | Smaller in Size | Charge on Wild Type | Charge on Mutant | Domain |
| C321R | Wild Type | Wild Type | Neutral | Positive | Tyrosinase_Cu-bd |
| | | | | | Common central domain of tyrosinase |
| I151N | Wild Type | Wild Type | None | None | NA |
| K142M | Mutant | Mutant | Positive | Neutral | NA |
| L189P | None | Wild Type | None | None | Tyrosinase_Cu-bd |
| | | | | | Common central domain of tyrosinase |
| M179R | Wild Type | Wild Type | Neutral | Positive | Tyrosinase_Cu-bd |
| | | | | | Common central domain of tyrosinase |
| S184L | Mutant | Wild Type | None | None | Tyrosinase_Cu-bd |
| | | | | | Common central domain of tyrosinase |

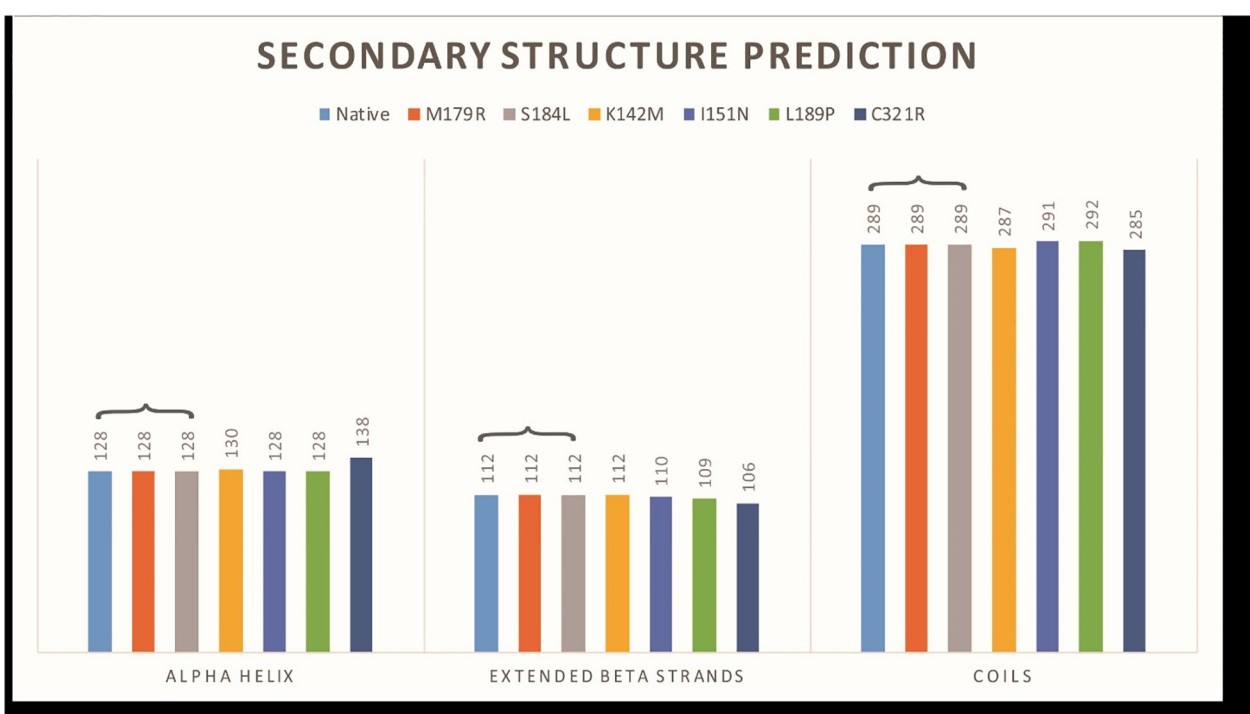

**Fig 3. Secondary structure prediction of wild type and all mutants.** Models are classified according to secondary structures. Three models; Native, M179R, and S184L have same structure.

docked complexes. Interestingly, mutant models L189P and C321R in docked complexes exhibited a lack of interaction between binding site residues and ligands. These findings enhance our understanding of the dynamic interactions within the active site and illuminate potential therapeutic ways for the identified tyrosinase protein mutations.

## Discussion

Mutations in the tyrosinase protein may disrupt the normal melanin production pathway by inducing abnormal production of 3,4-dihydroxyphenylalanine (DOPA), which subsequently converts into dopaquinone—a crucial molecule responsible for melanin pigment production [10, 35]. In addition, reduced melanin production diminishes skin protection against sunlight, elevating the risks of discoloration, uneven texture, and even skin cancer [36]. Conversely, heightened melanin production may lead to dark spots on the skin and adrenal disorders [37]. The over-expression of tyrosinase results in an increased intracellular dopamine content, coupled with formation of melanin pigments in neuronal somata, ultimately triggering apoptotic cell death [19]. Given tyrosinase's crucial role in the initial and rate-determining step of melanin production, any mutation in this protein can profoundly alter its function within the body [1, 4, 38]. Therefore, the use of advanced bioinformatics tools and computational approaches [34] to investigate the structural and functional impact of disease-causing nsSNPs on tyrosinase protein helped in studying the alterations in the normal functioning of tyrosinase protein. Sequence-based strategies provided a great podium to study closely associated protein members [30].

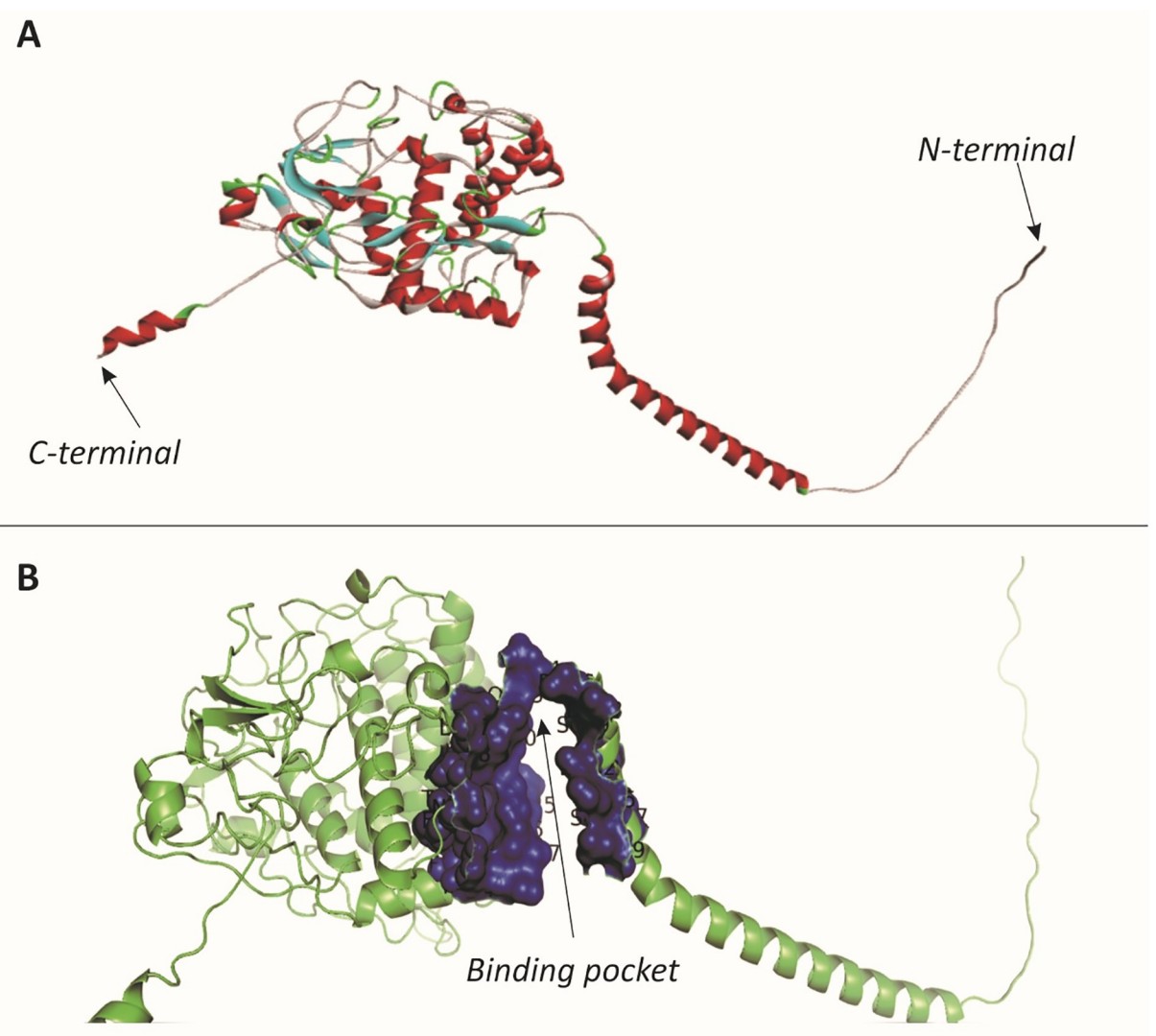

**Fig 4.** A) Protein model of Tyrosinase protein. C-terminal and N-terminal end of the 3D structure ended with coil and loop respectively. B). Binding pocket of Tyrosinase protein is shown in blue color.

In this study, we identified six particularly detrimental variants—K142M, I151N, M179R, S184L, L189P, and C321R—that significantly compromise protein stability. It is also crucial to note that all six mutations were situated within the highly conserved and functional region of the tyrosinase protein. Previous studies indicated the existence of a primary domain, referred to as "Tyrosinase_Cu-bd," which serves as a common central domain of tyrosinase [39, 40]. This domain was found to harbor four mutations—C321R, L189P, S184L, and M179R. Notably, two mutations, I151N and K142M, were not confined within the tyrosinase_Cu-bd central domain. These structural changes in terms of hydrophobicity, size, and charge on the mutant and wild-type protein structure may associate with the disrupt of the normal melanin production.

Molecular docking analysis [32] revealed the structural alteration in mutant models as compared to the wild-type protein structure. In this study, 10 FDA approved drugs (Sapropterin, Azelaic Acid, Menobenzone, Levodopda, Mequinol, Arbutin, Hexylresorcinol, Artenimol,

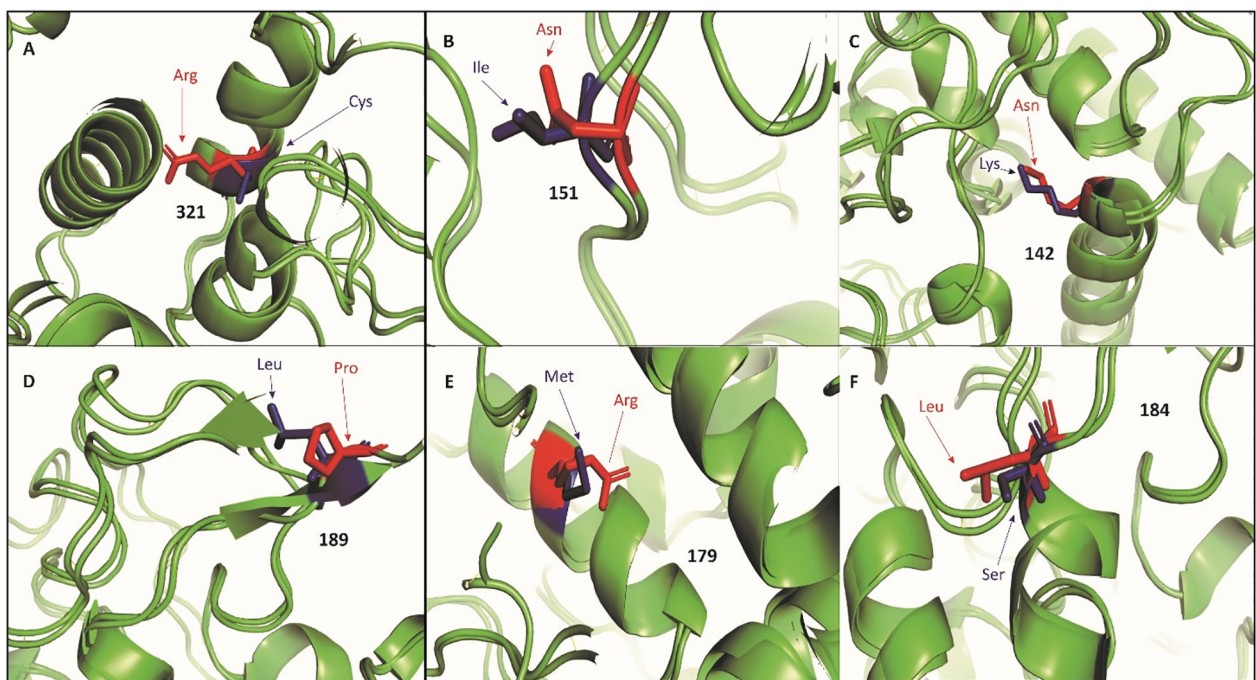

**Fig 5. Superimposed structures of deleterious variants C321R(5a), I151N(5b), K142M(5c), L189P(5d), M179R(5e), and S184L(5f).** All the wild type and mutant residues are shown in blue and red colors respectively.

Alloin, Curcumin) were used as a target and/or inhibitor of tyrosinase protein with wild type and six mutant models of tyrosinase protein. The results (Table 3) depicted that 15 residues (117Leu,118Leu, 119Val, 121Arg, 124Phe, 126Leu, 130Glu, 133Lys, 134Phe, 230Asn, 231Phe, 232Thr, 234Pro, 453Gln, 454Asp) from the binding pocket of tyrosinase protein were involved in the protein-ligands interactions. The native model did not interact with ligands through a binding pocket proving that these ligands do not bind with the normal tyrosinase protein. However, some ligands interact with the binding sites in four mutant models K142M, I151N, M179R, and S184L proving that these ligands directly bind with the active site of mutant tyrosinase protein to inhibit it's working. Two mutant models L189P and C321R on the other hand did not show any binding site residues interaction with any ligands. The interacting residues of these two models with all 10 ligands showed a very different residual interaction as compared to the native and other four mutant models' docking interactions. Similarly, some researchers assessed potential pharmaceutical agents targeting both wild-type ABL1 and the T315I mutant ABL1. They conducted molecular docking analyses involving some FDA-approved small-molecule drugs. They revealed chlorhexidine and sorafenib as promising "new use" drugs specifically targeting wild-type ABL1, while nicergoline and plerixafor exhibited potential for targeting T315I ABL1. Notably, their investigation highlighted the pivotal involvement of residues situated within the ATP-binding site and the A-loop motif in the drug discovery process targeting ABL1 [41]. In another study, Banavath et al. identified novel tyrosine kinase inhibitors for drug resistant T315I mutant BCR-ABL via molecular docking [42]. Others have done similar type of studies on various targets [43–46]. Therefore, our study suggests that clinical trials and in vitro experiments validating these findings hold significant promise in advancing the fight against skin cancer and associated diseases.

**Table 3. Interaction analysis of FDA approved (Drugs) ligands with wild-type and mutant structures of tyrosinase.**

| Name of Ligand | Name of Receptor | Conventional Hydrogen Bonds | Van der Waals Interactions | Carbon Hydrogen Bonds | Pi-Bonds | Unfavorable Bonds |
|---|---|---|---|---|---|---|
| **Sapropterin** | Native | Cys103, Lys104, Phe107, Arg116 Gln223, Gly446, | Met96, Asn102 Phe105, Gly106, Trp108, Leu216, Leu118, Glu219, Gln220, Glu229, Leu445, Tyr447 | | Cys103 | Cys103 |
| | K142M | Arg121, Glu130, Asn230 | Leu118, Val119, Leu126, Lys133, Phe134, Phe231, Pro234 | | Thr232 | |
| | I151N | CYS89, GLY97 | LEU59, THR88, GLN90, CYS91, PHE98, CYS100, TYR425, ARG434, ASP437 | LEU432 | MET96, TYR433, PRO431 | |
| | M179R | Glu229, Arg116, Gly446 | Asn102, Phe107, Gln220, Phe105, Leu445, Tyr447, Leu118, Gln223, Trp108, Lys224, Gly109 | Lys104, Cys103, Gly106 | Arg116 | |
| | S184L | Ser375 | His367, Met374, Gln376, Ser380, Phe386, Leu184, Phe207, Ile368, Ser360 | Asn364, His202 | Val377, Phe347 | His180 |
| | L189P | Asn102, Asp437 | Phe429, Cys103, Arg217, Gly97, Leu213, Gln220, Leu445, Ile440 | | Met96, Phe105, Phe438, Tyr433, Lys104 | |
| | C321 | CYS89, ASP437 | LEU59, THR88, GLN90, CYS91, GLY97, TYR425, PRO431, LEU432, ARG434, | | MET96, TYR433 | |
| **Azelaic Acid** | Native | Phe105 | Phe107, Gly106, Asp444, Leu445, Phe438, Leu432, Asp437, Arg434 | | Tyr433, Lys104, Met96 | |
| | K142M | Ser79, Trp80 | Ser61, Asn62, Ala63, Pro64, Thr72, Arg77, Glu78, Phe200, Gly419, Asn421, Ser424, Tyr425, Val427 | | Leu65, Ile418 | |
| | I151N | ARG217 | GLY97, PHE98, ASN102, GLN220, PRO431, LEU445 | | MET96, PHE105, LEU213, PHE429, TYR433, PHE438 | ASP437 |
| | M179R | | Asn230, Pro234, Glu130, Phe134, Tyr137 | | Lys133, Phe231, Leu126 | Thr232, Asp228 |
| | S184L | Glu130, Arg121 | Phe134, Thr232, Lys133, Tyr137, Asp228 | | Pro234, Leu126, Phe231 | |
| | L189P | Arg217, Asn102, Cys103 | Gly97, Lys104, Gln220, Leu445, Asp437, Phe438 | | Phe429, Leu213, Met96, Tyr433, Phe105, | |
| | C321 | TRP80, TYR425 | LEU60, SER61, ALA63, PRO64, THR72, GLU78, SER79, GLY419, SER424 | | LEU65, ILE418 | ASN62 |
| **Menobenzone** | Native | | Met96, Phe438, Phe105, Ile440, Asp437, Leu432 | | Lys104, Leu445, Tyr433 | |
| | K142M | His202 | His180, Ser360, His363, His367, Ser375, Phe386, His390 | Asn364 | Val377, Ile368 | Ser380 |
| | I151N | | Leu117, Leu118, Arg121, Glu130, Lys133, Phe134, Asn230, Phe231, Pro234 | | Thr232, Leu126 | Val119 |
| | M179R | | Phe134, Lys133, Arg121, Asp228, Asn230, Thr232, Phe231, Pro234 | Glu130 | Leu126 | |
| | S184L | His202 | His363, His180, Ser375, Ser360, Phe386, His390, His367, Phe347 | Asn364 | Val377,Ile368 | Ser380 |
| | L189P | | Asn102, Arg217, Cys103, Lys104, Leu445, Asp437, Leu213, Phe429, Pro431, Gly97, Gln220 | | Tyr433, Phe438, Phe105, Met96 | |
| | C321 | SER79 | SER61, ALA63, GLY419, HIS420, ASN421, SER424 | GLU78 | THR72, LEU65, ILE418 | |

*(Continued)*

**Table 3.** (Continued)

| Name of Ligand | Name of Receptor | Conventional Hydrogen Bonds | Van der Waals Interactions | Carbon Hydrogen Bonds | Pi-Bonds | Unfavorable Bonds |
|---|---|---|---|---|---|---|
| **Levodopda** | Native | Leu432, Cys91 | Gln90, Leu59, Asn57, Ile58, Tyr425, Pro431, Thr88, Cys89 | | | |
| | K142M | Ser61, Asn62, Ala63, Ser79 | Pro64, Thr72, Glu78, Trp80, Gly419, His420, Asn421, Ser424 | | Ile418, leu65 | |
| | I151N | SER61, ASN62, ALA63, SER424, TYR425 | LEU60, PRO64, LEU65, ARG77, GLU78, SER79, PHE200, ILE418, GLY419, ASN421, MET426, VAL427 | TRP80, SER424 | | |
| | M179R | Glu130 | Lys133, Asn230, Val119, Thr232, Pro234, Phe231, Phe134 | | Leu126 | Arg121 |
| | S184L | Glu130, Arg121, Val119 | Lys133, Phe231, Phe134, Tyr137, Pro234, Asn230, Thr232 | | Leu126 | |
| | L189P | Asn102, Cys103, Arg217 | Asp437, Leu445, Lys104, Phe429, Gln220, Gly97, Leu213 | Phe438 | Met96, Tyr433, Phe105 | |
| | C321 | ALA63, ARG77, SER79, SER424, TYR425 | SER61, ASN62, PRO64, LEU65, THR72, GLU78, TRP80, PHE200, ILE418, GLY419, ASN421, VAL427 | | | |
| **Mequinol** | Native | Arg116 | Gln220, Gln223, Cys103, Phe105, Leu216, Tyr447, Gly446, | Gly106, Phe107 | Leu118 | Leu445 |
| | K142M | Ser79 | Ala63, Thr72, Pro417, His420, Asn421, Ser424 | Glu78 | Leu65, Ile418 | Gly419 |
| | I151N | | Arg121, Lys133, Phe134, TYR137, Phe231, Thr232 | Glu130 | Leu126, Pro234 | |
| | M179R | Ser79 | Gly419, Ser424, His420, Asn421, Ala63, Thr72, Glu78 | | Ile418, Leu65 | |
| | S184L | Ser79 | Thr72, Ala63, His420, Asn421, Ser424, Ser61, Gly419, | Glu78 | Ile418, Leu65 | |
| | L189P | | Gly97, Tyr149, Arg217, Asn102, Gln220, Arg116, Cys103, Lys104 | Ile430 | Phe105, Met96, Phe429, Tyr433, Pro431 | |
| | C321 | SER79 | SER61, ALA63, THR72, GLY419, HIS420, ASN421, SER424 | GLU78 | ILE418, LEU65 | |
| **Arbutin** | Native | Asp437, Leu59, Asn57, Cys89, Cys91 | Arg434, Tyr433, Ile58, Leu432, Thr88, Gln90, Pro431, Gly97, Met96, Phe95, Lys104 | | Tyr425 | |
| | K142M | Arg196, Ile198, Asn364 | Ser184, Met185, Asp186, Asp197, Asp199, His202, Phe347, Gln378 | | Glu203, Val377 | |
| | I151N | Glu130 | Leu118, Val119, Phe134, TYR137, Asn230, Phe231, Thr232 | | Lys133 | Arg121 |
| | M179R | Tyr282 | Gly191, Gly190, Thr155, Ser287, Ser284, Glu281, Asn283, His285, Leu188 | His285, Ser287 | | |
| | S184L | Ser375, Asn364 | Asp199, His202, Ser360, Ala365, Ile368. His367, Ser380, Phe347 | | Glu203, Val377 | |
| | L189P | Arg217, Cys103, Asp437 | Leu445, Leu213, Lys104, Gln220, Asn102, Ile430, Gly97, Tyr149, Pro431, Phe429 | | Phe105, Tyr433, Phe438, Met96 | Phe98 |
| | C321 | ARG196, ILE198, ASN364 | SER184, MET185, ASP186, ASP197, ASP199, HIS202, PHE347, GLN378 | | GLU203, VAL377 | |

(Continued)

**Table 3.** (Continued)

| Name of Ligand | Name of Receptor | Conventional Hydrogen Bonds | Van der Waals Interactions | Carbon Hydrogen Bonds | Pi-Bonds | Unfavorable Bonds |
|---|---|---|---|---|---|---|
| Hexylresorcinol | Native | Gln90 | Tyr425, Leu59, Ile58, Thr88, Cys89, Pro431, Met96, Lys104 | | Leu432, Cys91, Tyr433 | |
| | K142M | | His180, His202, Phe207, Phe347, Ser360, His363, Asn364, His367, Ser375, Gln376, Ser380, Phe386 | | Val377, Ile368 | |
| | I151N | Glu130 | Arg121, Phe134, ASP228, Asn230, Thr232 | | Leu126, Lys133, TYR137, Phe231, Pro234 | |
| | M179R | | Glu130, Lys133, Tyr137, Thr232, Asn230, Asp228 | Phe231 | Phe134, Pro234, Leu126, Phe231 | |
| | S184L | Val377 | Asn364, Phe347, His202, His180, Leu184, Ser380, Phe386, Gln376, Ser375, His367, Thr309, Ser360 | | Val377, Ile368 | |
| | L189P | Arg217 | Phe438, Pro431, Gly97, Asn102, Gln220, Cys103, Asp437 | | Leu445, Met96, Tyr433, Leu213, Phe429, Lys104, Phe105 | |
| | C321 | | LYS131, ASP132, ALA136, GLU250, TYR251, MET252, GLY253, SER267 | | PHE135, PHE268 | |
| Artenimol | Native | Glu229 | Gly227, Lys224, Gln223, Gln220, Arg116, Trp108, Phe107, Cys103, Gly109, Gly101, Asn102 | | | |
| | K142M | Lys133 | Val119, Arg121, Leu126, Glu130, Phe134, Asp228, Asn230, Thr232, Pro234 | Phe231 | | |
| | I151N | Lys133 | Val119, Arg121, Leu126, Glu130, Phe134, ASP228, Asn230, Thr232, Pro234 | Phe231 | | |
| | M179R | Gly66, Asn421 | Pro417, Met332, His420, Pro64, Ala414, Asn415, Leu65, Ile418 | Pro417, Asn415 | Pro205, | |
| | S184L | | Glu203, His202, Phe207, Phe347, His367, His363, Asn364, Ser360, Ile368 | | Val377 | |
| | L189P | | Gln378, Ser184, Arg196, Ile198, Glu203, Asp197, His202, Phe347 | | Val377 | Asp199 |
| | C321 | | ASN122, ASP237, ASP240, CYS247 | | ILE246, Phe124 | TRP236 |
| Alloin | Native | Leu452, Arg402, Arg403 | Gln399, Arg405, Asp454 | | Arg402, Arg403 | |
| | K142M | Val119, Arg121, Glu130, Phe134 | Lys133, Tyr137, Asn230, Phe231, Thr232, Pro234 | | Glu130, Leu126 | Asp228 |
| | I151N | LEU188, GLN286 | ASP75, VAL83, THR155, TYR156, GLN158, ASN161, LEU189, GLY190 | GLY73, VAL74 | ASP76 | GLY157, LYS160 |
| | M179R | Asn364, Gln359, Ala357 | Ser360, Arg308, Thr309, Gln376, Ser375, Val377, Ser349, Gly346, Phe347, Lys334 | | Ile368 | Ala348, Ser358 |
| | S184L | | Arg121, Val119, Leu118, Leu117, Thr232, Pro234, Phe134, Lys133, Phe231, Tyr137 | Asn230 | Asp228, Leu126 | Glu130 |
| | L189P | Asp199, Glu345, Gln376 | Gln378, Ser375, Ile368, His363, His367, Phe347, His202, Glu203 | Asn364 | Val377 | |
| | C321 | GLU203, HIS367 | HIS202, THR309, LYS334, SER360, HIS363, ASN364, ALA365, SER375 | | PHE347, ILE368, VAL377 | |

(*Continued*)

**Table 3.** (Continued)

| Name of Ligand | Name of Receptor | Conventional Hydrogen Bonds | Van der Waals Interactions | Carbon Hydrogen Bonds | Pi-Bonds | Unfavorable Bonds |
|---|---|---|---|---|---|---|
| Curcumin | Native | Cys103 | Gly227, Glu229, Trp108, Gly109, Asn102, Phe107, Gln220, Gly446, Leu445, Tyr447 | Leu216, Gln223, Glu219, Gly106 | Lys224, Arg116, Leu118, Phe105 | Lys224 |
| | K142M | Arg121, Lys133 | Val119, Ile123, Phe134, Tyr137, Asp228, Asn230, Phe231, Thr232, Pro234 | Glu130 | Leu126 | |
| | I151N | LEU279 | LEU188, ILE194, ARG196, GLU280, GLU281, TYR282, SER284, HIS285, GLN286 | TYR282, SER284 | | |
| | M179R | Arg52 | Pro110, Phe95, Gly101, Leu49, Ser40, Gly47, Asn99, Asn102, Phe107, Gln220, Trp108, Lys224, Glu229 | Gly109, Ser44 | Arg52, Trp39, Cys103, | |
| | S184L | Arg52, Glu229 | Ser44, Gly47, Leu49, Ser50, Phe95, Asn99, Gly101, Asn102, Phe107, Trp108, Pro110, Arg116, Gln220, | Gly109, | Lys224, Arg52, Trp39, Cys103, Trp39 | |
| | L189P | His367, Asn364 | Gln378, Asp199, Phe347, His363, Gln376, Arg196, Ser184, Ile198, Glu203, Ser380, Phe207 | | Val377, His202, His180, His202, His367, Val377 | |
| | C321 | LYS334, SER360 | ASP199, HIS202, THR309, ALA357, ASN364, ALA365, HIS367,ILE368, SER375 | GLU203 | VAL377, PHE347 | |

## Conclusion

In conclusion, our study identified significant non-synonymous single nucleotide polymorphisms (nsSNPs) in the TYR gene, particularly K142M, I151N, M179R, S184L, L189P, and C321R, which adversely impacted the stability of the tyrosinase protein. Additionally, we explored the potential inhibitory effects of 10 FDA-approved drugs on mutated tyrosinase structures. Notably, these drugs exhibited binding interactions with specific mutant models, providing valuable insights for targeted pharmaceutical interventions in the intricate network of melanin biosynthesis. This comprehensive approach enhances understanding of molecular variations and suggests promising ways for further research and therapeutic development for discoloration, uneven texture, skin cancer, dark spots on the skin, and adrenal disorders.

## Supporting information

**S1 Table. Reported the most deleterious variants decreasing the protein stability while present in the highly conserved region of the tyrosinase protein.**
(DOCX)

## Author Contributions

**Conceptualization:** Wei Fan.

**Data curation:** Heng Li Ji.

**Investigation:** Heng Li Ji.

**Methodology:** Shabbir Ahmed.

**Resources:** Mohibullah Kakar, Hussah M. Alobaid, Yasmeen Shakir.

**Software:** Shabbir Ahmed.

**Supervision:** Yasmeen Shakir.

**Validation:** Mohibullah Kakar.

**Visualization:** Hussah M. Alobaid.

**Writing – review & editing:** Hussah M. Alobaid, Yasmeen Shakir.

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
