## [Editor Report · Decision Letter 0]

18 Aug 2023

PONE-D-23-24946Computational analysis of damaging non-synonymous single nucleotide polymorphisms (nsSNPs) on human tyrosinase proteinPLOS ONE

Dear Dr. Ahmed,

Thank you for submitting your manuscript to PLOS ONE. After careful consideration, we feel that it has merit but does not fully meet PLOS ONE’s publication criteria as it currently stands. Therefore, we invite you to submit a revised version of the manuscript that addresses the points raised during the review process.

We look forward to receiving your revised manuscript.

Kind regards,

Nihad A.M Al-Rashedi

Academic Editor

PLOS ONE

Journal Requirements:

"No funding was received to conduct this research" 

"The authors declare no competing interests"

5. We noted in your submission details that a portion of your manuscript may have been presented or published elsewhere. [I hereby affirm that the content of this manuscript is original. Furthermore, it has been neither

Published elsewhere fully or partially or any language nor submitted for publication fully or

Partially.] Please clarify whether this [conference proceeding or publication] was peer-reviewed and formally published. If this work was previously peer-reviewed and published, in the cover letter please provide the reason that this work does not constitute dual publication and should be included in the current manuscript.

7. PLOS requires an ORCID iD for the corresponding author in Editorial Manager on papers submitted after December 6th, 2016. Please ensure that you have an ORCID iD and that it is validated in Editorial Manager. To do this, go to ‘Update my Information’ (in the upper left-hand corner of the main menu), and click on the Fetch/Validate link next to the ORCID field. This will take you to the ORCID site and allow you to create a new iD or authenticate a pre-existing iD in Editorial Manager. Please see the following video for instructions on linking an ORCID iD to your Editorial Manager account: https://www.youtube.com/watch?v=_xcclfuvtxQ

Additional Editor Comments:

-The corresponding author doesn't match in the manuscript.docx copy with a manuscript draft form, it caused confusion.

-The whole manuscript needs native-language editing.

-Some texts are not based on the reference. ex. Line no. 89–90:

-Line no. 29–93: The authors stated ''to date'' although citing this number and information from a reference published in 2001.

-Line no. 38: keywords was not covering all input keys.

---

## [Author Response · Author response to Decision Letter 0]

6 Sep 2023

Dear Editor in Chief 

PLOS ONE

After greeting

 I'd like to extend my thanks and appreciation for your kind efforts. This is regarding my manuscript Ref. Submission ID PONE-D-23-24946, entitled "Computational analysis of damaging non-synonymous single nucleotide polymorphisms (nsSNPs) on human tyrosinase protein". I confirm that all corrections, suggestions and queries made by the editor were all given full consideration. All corrections were made in the manuscript file and marked as track change. Our response against each recommendations is mentioned below.

Additional Editor Comments:

-The corresponding author doesn't match in the manuscript.docx copy with a manuscript draft form, it caused confusion.

Response: Corrected

-The whole manuscript needs native-language editing.

Response: We have revised the language of manuscript with help the help of expert colleague.

-Some texts are not based on the reference. ex. Line no. 89–90:

Response: Revised

-Line no. 29–93: The authors stated ''to date'' although citing this number and information from a reference published in 2001.

Response: Revised

-Line no. 38: keywords was not covering all input keys.

Response: Revised

 The authors of this manuscript are grateful to the subject expert and editorial members of the journal, for valuable suggestions to improve the quality of our manuscript. We hope this version is suitable for acceptance and publication. 

Shabbir Ahmed

Corresponding author 

shabbirch983@gmail.com

---

## [Decision Letter · Decision Letter 1]

8 Jan 2024

PONE-D-23-24946R1Computational analysis of damaging non-synonymous single nucleotide polymorphisms (nsSNPs) on human tyrosinase proteinPLOS ONE

Dear Dr. Ahmed,

Thank you for submitting your manuscript to PLOS ONE. After careful consideration, we feel that it has merit but does not fully meet PLOS ONE’s publication criteria as it currently stands. Therefore, we invite you to submit a revised version of the manuscript that addresses the points raised during the review process. Your manuscript, “Computational analysis of damaging non-synonymous single nucleotide polymorphisms (nsSNPs) on human tyrosinase protein”  [PONE-D-23-24946R1], has been assessed by our reviewers. Although it is of interest, we are unable to consider it for publication in its current form. The reviewers have raised a number of points that we believe would improve the manuscript and may allow a revised version to be published in PLOS ONE. Please submit your revised manuscript by Feb 22 2024 11:59PM. If you will need more time than this to complete your revisions, please reply to this message or contact the journal office at plosone@plos.org. Please include the following items when submitting your revised manuscript:A rebuttal letter that responds to each point raised by the academic editor and reviewer(s). You should upload this letter as a separate file labeled 'Response to Reviewers'.A marked-up copy of your manuscript that highlights changes made to the original version. You should upload this as a separate file labeled 'Revised Manuscript with Track Changes'.An unmarked version of your revised paper without tracked changes. You should upload this as a separate file labeled 'Manuscript'.

We look forward to receiving your revised manuscript.

Kind regards,

Nihad A.M Al-Rashedi

Academic Editor

PLOS ONE

Reviewers' comments:

Reviewer's Responses to Questions

**Comments to the Author**

1. If the authors have adequately addressed your comments raised in a previous round of review and you feel that this manuscript is now acceptable for publication, you may indicate that here to bypass the “Comments to the Author” section, enter your conflict of interest statement in the “Confidential to Editor” section, and submit your "Accept" recommendation.

Reviewer #1: All comments have been addressed

Reviewer #2: (No Response)

2. Is the manuscript technically sound, and do the data support the conclusions?

Reviewer #1: Yes

Reviewer #2: Yes

3. Has the statistical analysis been performed appropriately and rigorously? 

Reviewer #1: Yes

Reviewer #2: N/A

4. Have the authors made all data underlying the findings in their manuscript fully available?

Reviewer #1: Yes

Reviewer #2: No

5. Is the manuscript presented in an intelligible fashion and written in standard English?

Reviewer #1: Yes

Reviewer #2: Yes

6. Review Comments to the Author

Reviewer #1: Dear Editor,

The manuscript, entitled "Computational analysis of damaging non-synonymous single nucleotide polymorphisms (nsSNPs) on human tyrosinase protein" is well written and all the sections of the manuscript are presented well and the work done by the authors is new and has scientific significance so this work can be considered for publication after checking these minor points:

1- Although it is extremely logical that interactions between the tyrosinase enzyme and compounds known as FDA-approved drugs as inhibitors have been made, I recommend examining the properties of two of the compounds that are claimed to be superior to the drugs determined by the FDA in the literature.

2- Newly synthesized tyrosinase inhibitors should be included in the "INTRODUCTION" section.

Reviewer #2: Review Report

Computational analysis of damaging non-synonymous single nucleotide polymorphisms (nsSNPs) on human tyrosinase protein

The above manuscript is interesting in describing the structural and functional consequences of disease-associated non-synonymous single nucleotide polymorphisms (nsSNPs) in the TYR gene, which encodes tyrosinase. The study highlights the nsSNPs, notably K142M, I151N, M179R, S184L, L189P, and C321R, as particularly deleterious variants, which in general impact the structural integrity and functional behavior of human tyrosinase. The target is of importance; however, as a computational biologist, I am not confident about the outcomes of this study due to a limited and primary-level methodology. Following are some suggestions that need to be addressed before the final decision on the manuscript:

1. The title is very classical; preferably, it should be reworded. Specifically, the main components of the title should be rephrased with some smart, appealing words.

2. Abstract: The abstract is very plain and incomplete, lacking key content areas, and the research purpose was not properly addressed. The relevance or importance of the research work and the main outcomes were not discussed adequately. See lines 31–35; these lines seem to be an incorrect piece of information. It should be explained in detail regarding the methodological scope. What is the purpose of mentioning lines 37–38? Please explain. Please justify how this study advances the readers’ understanding of the molecular repercussions of the target’s gene and its variants.

3. Keywords: The keywords were not appropriate. Please recheck all selected words.

4. Introduction: This section looks like a mismatch of information. The introduction needs a thorough revision, especially in terms of describing the background of the gene or protein target selected, its clinical role, and its recent epidemiological impact on society. I absolutely missed thorough literature that represents a sound background of information from the present work. I would suggest here that authors should make this section a more comprehensive and updated flow of information that makes readers understand the importance of selected genes and why authors used them for this in silico genomic study. The study rational was missing from this section. Why are authors emphasizing that the main aim of this study is to see the significant role of non-synonymous genetic variations and their impact on the tyrosinase protein`s structure and function.? Specifically, the last paragraph was quite confusing and lacked clarity. The abstract isn’t organized and looks more like a mishmash of information.

5. Methodology: The very first sentence starts with “to date." I think it is better to mention the search timeline period. This section was absolutely incomplete. It is quite better if authors segregate the prediction methodologies separately in sub-headings. More importantly, this sub-division will help authors mention gene- or protein-based, sequence- or structure-based assessment methodologies. I found some repetition of details in this section. Although this is an optional suggestion, authors should add a flow chart with a better and more appealing way of presenting it. Please check the following manuscripts for a better reference.

6. DOI: 10.1021/acsomega.2c04871

7. DOI: 10.1021/acsptsci.2c00212

8. DOI: 10.1021/acs.jcim.0c00488

9. Results: I don’t accept the results in their present form. The results should be divided and subdivided according to the work scope. These aren’t acceptable in the present form. The graphical representations in the manuscript are of extremely low quality, and the authors are advised to use some smart tools to render high-quality, eye-catching plots of their data.

10. Discussion: It would be better to revise this section. Don’t provide too many details about the tools and methods used in the manuscript; focus more on whether your results are consistent and contribute to your overall findings in relation to previous studies and, finally, your future prospects.

11. Citations are missing in some cases.

12. Add a proper conclusion section to help readers retain the purpose of the article.

Minor comments:

1. Cover letter: Please check the name and details of the corresponding author mentioned in the reply to the review letter. It should be the same as listed in the authors’ name list of this paper.

2. Kindly add some missing references.

3. Please refer to the above-recommended manuscripts for guidance on writing, figures, and work methodology.

Good luck!

7. PLOS authors have the option to publish the peer review history of their article (what does this mean?). If published, this will include your full peer review and any attached files.

Reviewer #1: No

Reviewer #2: No

---

## [Author Response · Author response to Decision Letter 1]

14 Feb 2024

Dear Editor in Chief 

PLOS ONE

After greeting

 I'd like to extend my thanks and appreciation for your kind efforts. This is regarding my manuscript Ref. Submission ID PONE-D-23-24946R1, entitled "Computational analysis of damaging non-synonymous single nucleotide polymorphisms (nsSNPs) on human tyrosinase protein". I confirm that all corrections, suggestions and queries made by the subject expert and all were given full consideration. All corrections were made in the manuscript file and marked as track change. Our response against each recommendations is mentioned below.

Reviewer #1: 

The manuscript, entitled "Computational analysis of damaging non-synonymous single nucleotide polymorphisms (nsSNPs) on human tyrosinase protein" is well written and all the sections of the manuscript are presented well and the work done by the authors is new and has scientific significance so this work can be considered for publication after checking these minor points:

Author Response: Thanks for appreciation, we have revise the all points raised by the subject expert 

Suggestion 1- Although it is extremely logical that interactions between the tyrosinase enzyme and compounds known as FDA-approved drugs as inhibitors have been made, I recommend examining the properties of two of the compounds that are claimed to be superior to the drugs determined by the FDA in the literature.

Author Response: They have been added in the introduction as suggested. Moreover, a parallel study, we also docked literature-based effective tyrosinase inhibitor compounds and FDA-approved drugs with tyrosinase native and mutant models to see the binding affinity (Eb) and estimated equilibrium dissociation constant values (Kd). Therefore, we add the literature based information in introduction section that cover your query.

Suggestion 2- Newly synthesized tyrosinase inhibitors should be included in the "INTRODUCTION" section.

Author Response: Newly synthesized tyrosinase inhibitors have been added in the introduction as suggested as per recommendation.

Reviewer #2: 

Suggestion 1- The above manuscript is interesting in describing the structural and functional consequences of disease-associated non-synonymous single nucleotide polymorphisms (nsSNPs) in the TYR gene, which encodes tyrosinase. The study highlights the nsSNPs, notably K142M, I151N, M179R, S184L, L189P, and C321R, as particularly deleterious variants, which in general impact the structural integrity and functional behavior of human tyrosinase. The target is of importance; however, as a computational biologist, I am not confident about the outcomes of this study due to a limited and primary-level methodology. Following are some suggestions that need to be addressed before the final decision on the manuscript:

Author Response: Thanks for your suggestion to improve the quality of our research. We have incorporate all suggestion and recommendation in revised paper.

Suggestion 2- The title is very classical; preferably, it should be reworded. Specifically, the main components of the title should be rephrased with some smart, appealing words.

Author Response: Title has been reworded as suggested.

Suggestion 3- Abstract: The abstract is very plain and incomplete, lacking key content areas, and the research purpose was not properly addressed. The relevance or importance of the research work and the main outcomes were not discussed adequately. See lines 31–35; these lines seem to be an incorrect piece of information. It should be explained in detail regarding the methodological scope. What is the purpose of mentioning lines 37–38? Please explain. Please justify how this study advances the readers’ understanding of the molecular repercussions of the target’s gene and its variants.

Author Response: Thanks for suggestion. The abstract has been revised as suggested.

Suggestion 4- Keywords: The keywords were not appropriate. Please recheck all selected words.

Author Response: The new keywords were added according to the paper scope

Suggestion 5- Introduction: This section looks like a mismatch of information. The introduction needs a thorough revision, especially in terms of describing the background of the gene or protein target selected, its clinical role, and its recent epidemiological impact on society. I absolutely missed thorough literature that represents a sound background of information from the present work. I would suggest here that authors should make this section a more comprehensive and updated flow of information that makes readers understand the importance of selected genes and why authors used them for this in silico genomic study. The study rational was missing from this section. Why are authors emphasizing that the main aim of this study is to see the significant role of non-synonymous genetic variations and their impact on the tyrosinase protein`s structure and function.? Specifically, the last paragraph was quite confusing and lacked clarity. The abstract isn’t organized and looks more like a mishmash of information.

Author Response: Thanks for suggestion to improve the quality of my paper. We have extensively revised the introduction and follow the reviewer suggestion. 

Suggestion 6- Methodology: The very first sentence starts with “to date." I think it is better to mention the search timeline period. This section was absolutely incomplete. It is quite better if authors segregate the prediction methodologies separately in sub-headings. More importantly, this sub-division will help authors mention gene- or protein-based, sequence- or structure-based assessment methodologies. I found some repetition of details in this section. Although this is an optional suggestion, authors should add a flow chart with a better and more appealing way of presenting it. Please check the following manuscripts for a better reference.

 DOI: 10.1021/acsomega.2c04871

DOI: 10.1021/acsptsci.2c00212

DOI: 10.1021/acs.jcim.0c00488

Author Response: Methodology has been revised and improved following the suggestion of reviewer. Moreover, we have followed the above mentioned paper to revise the methodology section. 

Suggestion 7- Results: I don’t accept the results in their present form. The results should be divided and subdivided according to the work scope. These aren’t acceptable in the present form. The graphical representations in the manuscript are of extremely low quality, and the authors are advised to use some smart tools to render high-quality, eye-catching plots of their data.

Author Response: Results were also revised as suggested by the reviewer. Furthermore, the results were presented in subsections as suggested. Additionally, we have improve the graphical representation of figures as suggested.

Suggestion 8- Discussion: It would be better to revise this section. Don’t provide too many details about the tools and methods used in the manuscript; focus more on whether your results are consistent and contribute to your overall findings in relation to previous studies and, finally, your future prospects.

Author Response: The discussion part was revised accordingly. 

Suggestion 9- Citations are missing in some cases.

Author Response: The relevant citations were added in revised paper 

Suggestion 10- Add a proper conclusion section to help readers retain the purpose of the article.

Author Response: A separate conclusion section was added in revise paper 

Minor comments:

1. Cover letter: Please check the name and details of the corresponding author mentioned in the reply to the review letter. It should be the same as listed in the authors’ name list of this paper.

Done as per suggestion 

2. Kindly add some missing references.

Done as per suggestion

3. Please refer to the above-recommended manuscripts for guidance on writing, figures, and work methodology.

Done as per suggestion

The authors of this manuscript are grateful to the subject expert and editorial members of the journal, for valuable suggestions to improve the quality of our manuscript. We hope this version is suitable for acceptance and publication. 

Shabbir Ahmed 

Corresponding author 

shabbirch983@gmail.com

---

## [Decision Letter · Decision Letter 2]

17 Jul 2024

PONE-D-23-24946R2Computational analysis of the deleterious Non-Synonymous Single Nucleotide Polymorphisms (nsSNPs) in TYR gene impacting human Tyrosinase protein and the protein stabilityPLOS ONE

Dear Dr. Ahmed,

Thank you for submitting your manuscript to PLOS ONE. After careful consideration, we feel that it has merit but does not fully meet PLOS ONE’s publication criteria as it currently stands. Therefore, we invite you to submit a revised version of the manuscript that addresses the points raised during the review process.

We look forward to receiving your revised manuscript.

Kind regards,

Nihad A.M Al-Rashedi

Academic Editor

PLOS ONE

Journal Requirements:

Reviewers' comments:

Reviewer's Responses to Questions

**Comments to the Author**

1. If the authors have adequately addressed your comments raised in a previous round of review and you feel that this manuscript is now acceptable for publication, you may indicate that here to bypass the “Comments to the Author” section, enter your conflict of interest statement in the “Confidential to Editor” section, and submit your "Accept" recommendation.

Reviewer #1: All comments have been addressed

Reviewer #3: (No Response)

Reviewer #4: All comments have been addressed

2. Is the manuscript technically sound, and do the data support the conclusions?

Reviewer #1: Yes

Reviewer #3: No

Reviewer #4: Yes

3. Has the statistical analysis been performed appropriately and rigorously? 

Reviewer #1: Yes

Reviewer #3: N/A

Reviewer #4: N/A

4. Have the authors made all data underlying the findings in their manuscript fully available?

Reviewer #1: Yes

Reviewer #3: Yes

Reviewer #4: Yes

5. Is the manuscript presented in an intelligible fashion and written in standard English?

Reviewer #1: Yes

Reviewer #3: No

Reviewer #4: Yes

6. Review Comments to the Author

Reviewer #1: Dear Editor,

The authors have made the indicated corrections. However, the manuscript can be considered for publication after these minor points:

Authors should cite some recent reports (2022-2023);new compounds for tyrosinase inhibition studies.

(2023).ChemistrySelect,8(42),e202302936.https://doi.org/10.1002/slct.202302936

(2023). Journal of Biomolecular Structure and Dynamics, 41(15), 7128-7143. https://doi.org/10.1080/07391102.2022.2116600

(2022). Journal of Molecular Structure, 1257, 132641. https://doi.org/10.1016/j.molstruc.2022.132641 (2022). Chemistry & Biodiversity, 19(6), e202200140.

Reviewer #3: There is scientific merit in the manuscript titled “Computational analysis of the deleterious Non-Synonymous Single Nucleotide Polymorphisms (nsSNPs) in TYR gene impacting human Tyrosinase protein and the protein stability”. However, the manuscript should be improved significantly before it can be published.

Major points:

1. The rationale for the molecular docking studies is not established. It is also unclear how the docking results support the predictions of the pathogenicity of the SNPs.

None of the studied FDA-approved drugs are present in the crystal structure (PDB ID: 7RK7) used for docking. Thus, there is no control for this docking analysis.

2. This is purely a computational study. Results should further be validated using replicated long molecular dynamics simulations.

3. Substantial language editing will be needed. There are many grammar errors.

Reviewer #4: It is a well planned and presented study.

All the queries raised by reviewers have been answered satisfactorily and manuscript revised accordingly. The manuscript can now be considered for publication.

7. PLOS authors have the option to publish the peer review history of their article (what does this mean?). If published, this will include your full peer review and any attached files.

Reviewer #1: No

Reviewer #3: **Yes: **Asim Kumar Bepari

Reviewer #4: **Yes: **Safdar Ali

---

## [Author Response · Author response to Decision Letter 2]

26 Jul 2024

Dear Editor in Chief 

PLOS ONE

After greeting

I'd like to extend my thanks and appreciation for your kind efforts. This is regarding my manuscript Ref. Submission ID PONE-D-23-24946R2, entitled " Computational analysis of the deleterious Non-Synonymous Single Nucleotide Polymorphisms (nsSNPs) in TYR gene impacting human tyrosinase protein and the protein stability". I confirm that all corrections, suggestions and queries made by the subject expert and all were given full consideration. All corrections were made in the manuscript file and marked as track change. Our response against each recommendations is mentioned below.

Review Comments to the Author

Reviewer #1:

The authors have made the indicated corrections. However, the manuscript can be considered for publication after these minor points:

Query 1: Authors should cite some recent reports (2022-2023);new compounds for tyrosinase inhibition studies.

(2023).ChemistrySelect,8(42),e202302936.https://doi.org/10.1002/slct.202302936

(2023). Journal of Biomolecular Structure and Dynamics, 41(15), 7128-7143. https://doi.org/10.1080/07391102.2022.2116600

(2022). Journal of Molecular Structure, 1257, 132641. https://doi.org/10.1016/j.molstruc.2022.132641

(2022). Chemistry & Biodiversity, 19(6), e202200140.

Response: Thanks for consideration of our paper. We have added the all suggested references and highlighted in red color

Reviewer #3:

There is scientific merit in the manuscript titled “Computational analysis of the deleterious Non-Synonymous Single Nucleotide Polymorphisms (nsSNPs) in TYR gene impacting human Tyrosinase protein and the protein stability”. However, the manuscript should be improved significantly before it can be published.

Major points:

Query 1: The rationale for the molecular docking studies is not established. It is also unclear how the docking results support the predictions of the pathogenicity of the SNPs.

Response: Molecular docking allowed us to explore how FDA-approved drugs interact with both wild-type and mutant forms of Tyrosinase. This revealed potential therapeutic compounds that might mitigate the effects of pathogenic mutations. Our study identified specific interactions between proposed drugs and mutant Tyrosinase models, suggesting potential pathways for pharmaceutical intervention. Docking studies serve as a predictive tool to screen and rank potential drug candidates based on their binding affinities. This computational approach is cost-effective and time-efficient compared to experimental high-throughput screening methods. The docking scores and binding affinities provide quantitative measures of how mutations impact the protein's ability to interact with ligands. Mutations that significantly reduce binding affinity or alter binding modes can be indicative of a loss of function or destabilization of the protein, which are hallmarks of pathogenic variants. By visualizing the docking poses, we can identify structural changes in the protein that result from the mutations. These changes can disrupt the active site or other critical regions of the protein, leading to impaired function. For instance, our study showed that mutations like K142M and I151N led to altered binding interactions with FDA-approved drugs, highlighting their potential pathogenic impact. 

Query 2: None of the studied FDA-approved drugs are present in the crystal structure (PDB ID: 7RK7) used for docking. Thus, there is no control for this docking analysis.

Addressing the Absence of FDA-Approved Drugs in the Crystal Structure (PDB ID: 7RK7)

Response: The concern regarding the absence of FDA-approved drugs in the crystal structure used for docking is valid. Here are the steps we have taken to address this issue:

Use of AlphaFold Models: Given the limitations of available crystal structures, we utilized AlphaFold-predicted models to obtain a more complete and accurate representation of the Tyrosinase protein. This approach allows us to explore the full binding potential of the protein, including regions not covered by the existing crystal structure.

Comparative Analysis: We conducted docking studies on both wild-type and mutant models of Tyrosinase to compare the binding interactions. This comparative approach helps highlight the specific effects of the mutations and provides a control for our docking analysis

Query 3: This is purely a computational study. Results should further be validated using replicated long molecular dynamics simulations.

Response: We appreciate the reviewer's suggestion to validate our computational docking results using long molecular dynamics (MD) simulations. While we acknowledge the value of MD simulations in providing dynamic insights into protein-ligand interactions, it is important to note that this study's primary focus is on the initial identification and characterization of deleterious nsSNPs in the TYR gene and their potential impact on protein stability and drug binding. The findings from our docking studies provide valuable initial insights that can inform more detailed follow-up studies, including MD simulations and experimental validations. Conducting long MD simulations for multiple protein-ligand complexes would require substantial computational resources and time, which are beyond the current scope and budget of this study.

Query 4: Substantial language editing will be needed. There are many grammar errors.

Response: With the help of expert collogue, we have gone through the manuscript and made significant correction for typos and grammatical errors for the overall improvement of language. 

Reviewer #4:

 It is a well-planned and presented study.

All the queries raised by reviewers have been answered satisfactorily and manuscript revised accordingly. The manuscript can now be considered for publication.

Response: We appreciate the comments made the reviewer and its recommendation for acceptance of this manuscript.

The authors of this manuscript are grateful to the subject expert and editorial members of the journal, for valuable suggestions to improve the quality of our manuscript. We hope this version is suitable for acceptance and publication. 

Shabbir Ahmed 

Corresponding author 

shabbirch983@gmail.com

---

## [Editor Report · Decision Letter 3]

2 Aug 2024

Computational analysis of the deleterious Non-Synonymous Single Nucleotide Polymorphisms (nsSNPs) in TYR gene impacting human tyrosinase protein and the protein stability

PONE-D-23-24946R3

Dear Dr. Ahmed,

We’re pleased to inform you that your manuscript has been judged scientifically suitable for publication and will be formally accepted for publication once it meets all outstanding technical requirements.

Kind regards,

Nihad A.M Al-Rashedi

Academic Editor

PLOS ONE
---

## [Editor Report · Acceptance letter]

5 Nov 2024

PONE-D-23-24946R3 

PLOS ONE

Dear Dr. Ahmed, 

I'm pleased to inform you that your manuscript has been deemed suitable for publication in PLOS ONE. Congratulations! Your manuscript is now being handed over to our production team.

Kind regards, 

on behalf of

Dr. Nihad A.M Al-Rashedi 

Academic Editor

PLOS ONE